# Developing a machine learning-based tool to extend the usability of the NICHD BPD Outcome Estimator to the Asian population

Monalisa Patel[1], Japmeet Sandhu[2], Fu-Sheng Chou[3]*

1 Division of Neonatology, Children's Hospital Orange County, Orange, California, United States of America, 2 Western University of Health Sciences, Pomona, California, United States of America, 3 Division of Neonatology, Department of Pediatrics, Loma Linda University School of Medicine, Loma Linda, California, United States of America

* FChou@llu.edu

**Data Availability Statement:** All relevant data are within the paper and its Supporting information files.

**Funding:** The authors received no specific funding for this work.

## Abstract

The NICHD BPD Outcome Estimator uses clinical and demographic data to stratify respiratory outcomes of extremely preterm infants by risk. However, the Estimator does not have an option in its pull-down menu for infants of Asian descent. We hypothesize that respiratory outcomes in extreme prematurity among various racial/ethnic groups are interconnected and therefore the Estimator can still be used to predict outcomes in infants of Asian descent. Our goal was to apply a machine learning approach to assess whether outcome prediction for infants of Asian descent is possible with information hidden in the prediction results using White, Black, and Hispanic racial/ethnic groups as surrogates. We used the three racial/ethnic options in the Estimator to obtain the probabilities of BPD outcomes for each severity category. We then combined the probability results and developed three respiratory outcome prediction models at various postmenstrual age (PMA) by a random forest algorithm. We showed satisfactory model performance, with receiver operating characteristics area under the curve of 0.934, 0.850, and 0.757 for respiratory outcomes at PMA 36, 37, and 40 weeks, respectively, in the testing data set. This study suggested an interrelationship among racial/ethnic groups for respiratory outcomes among extremely preterm infants and showed the feasibility of extending the use of the Estimator to the Asian population.

## Introduction

Bronchopulmonary dysplasia (BPD) is a multifactorial clinical syndrome of lung injury that disrupts the alveolarization and microvascular development [1]. Most infants with BPD have a prolonged need for respiratory support and supplemental oxygen. Despite advances in neonatology, a significant improvement in the incidence of BPD has not been observed, especially in extremely low gestational age infants. While these infants may eventually recover from respiratory support needs as the damage to the lung improves with adequate nutrition and growth over time, the diagnosis of BPD per se is also associated with an increased risk of long-term cardiopulmonary and neurodevelopmental impairment [2].

**Competing interests:** The authors have declared that no competing interests exist.

Preterm infants are at risk for mechanical, oxidant, and inflammatory injury because of lung underdevelopment and insufficient quantities of biochemical protectants, such as surfactant, antioxidants, and protease inhibitors [3]. The diagnosis of BPD is typically based on the NICHD 2001 workshop consensus. That is, a preterm infant must require supplemental oxygen for the first 28 postnatal days. The severity of BPD is further classified at 36 weeks postmenstrual age (PMA) based on the continuous need for respiratory support and the degree of $O_2$ supplementation that is needed. Severe BPD is defined as receiving supplemental $O_2$ of 30% or more, moderate BPD is defined as receiving 21–29% supplemental $O_2$, and mild BPD is defined as breathing room air by 36 weeks PMA [4].

Various interventions, including protective ventilation strategies, maintaining optimal oxygen saturation goals, timely surfactant administration, and the use of antenatal corticosteroids, have been shown to reduce BPD or risk factors associated with BPD [5]. Additionally, caffeine therapy for apnea of prematurity has also been linked to a decreased risk of BPD [6]. Vitamin A supplementation may also provide a modest benefit in oxygen dependency at 36 weeks PMA, likely related to its role in promoting the distal alveolar development [7, 8]. Postnatal corticosteroids have been used in clinical practice to decrease the length of invasive mechanical ventilation, a significant risk for BPD. However, its role in BPD prevention remains controversial. Moreover, postnatal corticosteroid use has been linked to risks of growth restriction, cardiac hypertrophy, and cerebral palsy [9, 10].

The use of these interventions is currently largely based on protocols of individual NICUs. In practice, neonatal providers would benefit from risk stratification to identify infants most likely to benefit from preventive therapies by using an objective risk estimating tool [11]. The BPD Outcome Estimator (the Estimator) is one of the available tools endorsed by the National Institute of Child Health and Human Development (NICHD) [12, 13]. The Estimator has been used to guide corticosteroid use and to counsel families, although it can be confounded by the varying rates of mortality among different racial/ethnic groups and the racial disparities in preterm birth [14–16]. In our institutional experience, one additional obstacle to the use of the Estimator was not being able to apply it to infants of Asian descent. At this time, due to limitations of the applicability of the Estimator, Asian families are being counseled with data that originally only applied to neonates of White, Black, or Hispanic descent. Therefore, we sought an approach to extend the usability of the Estimator to Asian extremely preterm infants. We hypothesize that respiratory outcomes in extreme prematurity among various racial/ethnic groups are interconnected and therefore the Estimator can be safely used to predict outcomes in infants of Asian descent. We aimed to test this hypothesis by first calculating the probability of each BPD severity group using the Estimator with demographic and respiratory data for Asian infants and created three sets of probability scores by choosing White, Black, and Hispanic as races/ethnicity for the same set of Asian infants. These scores were used as input into a machine learning algorithm for respiratory outcome prediction.

## Methods

### Patient population and data collection

We conducted a retrospective study at the Loma Linda University Children's Hospital (LLUCH) and Riverside University Health System (RUHS) neonatal intensive care units (NICUs). The Loma Linda University Health human research protection programs institutional review board (IRB) (#520338) and Riverside University Health System IRB (#1689889) both approved the study and both waived the consenting requirement given the retrospective nature of the study. Only the corresponding author (F.-S. C.) had access to the identifiable information and the raw data. The LLUCH NICU is a level 4 NICU and the RUHS NICU is a

level 3 NICU, both attended by the same group of neonatologists. Both NICUs have adapted to the same electronic medical record system since 2013. Our inclusion criteria included babies with gestational age (GA) of 30 weeks or less at birth, a birth weight of less than 1,250 grams, and a race/ethnicity designation as Asian in the admission face sheet in the electronic medical records from 2013 until 2020. A database search for patients that met inclusion criteria was performed by a data architect at LLU. Demographic as well as respiratory support, medications, and comorbidity data were extracted either directly from the database or manually via chart review. Babies that did not meet our inclusion criteria for gestational age and birth weight, as well as babies that did not have documented Asian race, were excluded. If the birth weight fell out of the range allowed in the Estimator based on the other demographic data, we used the closest weight to obtain probability scores.

## BPD risk estimation and primary outcomes

Demographic and respiratory data including GA, birth weight, sex, postnatal day, ventilator type, and $FiO_2$, on day of life 1, 3, 7, 14, 21, and 28 of the identified patients of Asian descent were entered into the Estimator to obtain probabilities of BPD for each severity category using White, Black, and Hispanic as surrogate race/ethnicity [12]. If more than one ventilator type was used within a day, the type that was used for the longest duration within the day, and the highest $FiO_2$ for that ventilator type were used. Three sets of probability score results were generated from each set of input data based on the three races/ethnicity entries. If the birth weight fell out of the range required based on the other parameters, the closest weight instructed by the Estimator was used for calculation. For ventilator type, non-invasive positive pressure ventilation (NIPPV) was considered "CPAP", and both high- and low-flow nasal cannulae were treated as "Cannula". Respiratory support needs at 36, 37, and 40 weeks PMA were used as outcomes for model development. Three models were developed for the prediction of the respiratory outcome at each PMA using the three sets of probability scores. For the 36-week PMA outcome designation, we classified the patients into two groups: one group with no respiratory support need or with low-flow nasal cannula (LFNC) use, and the other group with higher-level respiratory support needs; for the outcomes at 37 and 40 weeks PMA, we classified the patients into either with or without respiratory support needs. If a patient was discharged without respiratory support before the indicated PMA, the patient was classified into the group without support needs; on the other hand, if a patient was discharged before reaching the indicated PMA on respiratory support, the patient was classified into the group with support needs. None of the patients included in the study had a tracheostomy upon discharge, and none were discharged before 36 weeks PMA on LFNC. Similar to the analytic approach used in the BPD Outcome Estimator, we did not take repeated measures for each patient into consideration. In other words, each set of input data across various days of life for one patient was treated as an independent set of input data.

## Machine learning-based model development

Supervised machine learning was performed in this study. Specifically, the three sets of risk stratification probability results for the five severity categories (Death, Severe, Moderate, Mild, and No BPD) obtained from the three racial/ethnic surrogate groups were included as features (15 total independent variables, also known as features); respiratory support needs at either 36, 37, and 40 weeks PMA classified as mentioned above were included as the outcome for supervised learning. With the assumption that the probability scores for the five severity categories for each surrogate race/ethnicity already contained hidden information with regards to the demographic and respiratory support data in them, we did not incorporate the raw

demographic and respiratory support data in model development. Due to the intercorrelated nature among severity categories (e.g. higher probability in the Death or Severe categories negatively correlates with lower probability in the Mild or No BPD categories, or vice versa), non-linear algorithms were chosen. We tested random forest (RF) and kernel (radial) support vector machine (SVM) algorithms, as well as an ensemble algorithm that incorporated both RF and SVM.

Data were randomly split into two data sets in a 2:1 ratio, with the larger set used for training, and the small set for testing. A 10-fold cross-validation repeated 10 times was performed during training. For hyperparameter tuning, *mtry* was set as constant at 4; minimal node sizes of 1, 3, 5, and 10 were used, and splitting rules by the Gini index and extra trees were both tried for the final model selection.

For model performance assessment, metrics including sensitivity, specificity, positive and negative predictive values, overall predictive accuracy, and the *kappa* score were reported. The F1 score, calculated as the harmonic mean of sensitivity and positive predictive value, was also reported. A predicted probability of 50% or higher is considered as having a positive prediction. Additionally, the area under the receiver's operating characteristics curve (ROC AUC) was reported.

The study was performed in R 4.0.5 and RStudio 1.4 using the *caret* and *caretEnsemble* package [17, 18]. Codes are available upon request. We followed the TRIPOD guidelines checklist for prediction model development and validation [19].

## Results

A total of 829 preterm infants with a birth GA equal to or less than 30 weeks and a birth weight of less than 1,250 grams were identified in the clinical database. Three infants did not have race/ethnicity information. Forty-one infants were identified as Asian, among which 38 had complete respiratory data in the first 28 days of life. Demographic information is summarized in Table 1. The median GA was 26 weeks 3 days, with a range between 23 weeks 3 days to 30 weeks 6 days. The mean birth weight was 902 grams. Approximately 30% of the cohort were female. Among these newborns, 7 died and 2 were transferred out before reaching 36 weeks

**Table 1. Demographic characterization.**

| All infants identified as Asian (n = 38) | |
| --- | --- |
| Gestational age at birth | |
| *median* | 26 weeks 3 days |
| *range* | 23 weeks 3 days—30 weeks 6 days |
| Birth weight (gram) | |
| *Mean ± sd* | 902 ± 200 |
| Sex | |
| *Female* | 12 (31.6%) |
| *Male* | 26 (68.4%) |
| Birth place† | |
| *Inborn* | 31 (81.6%) |
| *Outborn* | 7 (18.4%) |
| Disposition | |
| *In-hospital mortality* | 7 (18.4%) |
| *Transferred before 36 weeks* | 2 (5.3%) |
| *Remainder (included in the study)* | 29 (76.3%) |

PMA. The remaining 29 infants had a total of 168 sets of input data available for model development and testing (Table 2). Table 2 shows their support modes at 36, 37, and 40 weeks. At 36 weeks' corrected gestational age (CGA), 27.6% (n = 8) patients did not require any support, 27.6% (n = 8) required respiratory support with a low-flow nasal cannula, and the remainder (44.8%, n = 13) required higher support. At 37 and 40 weeks' CGA, 34.5% (n = 10) and 51.7% (n = 15) patients did not require any support, respectively [20].

A total of 114 sets of input data were used for training, and the remaining 54 sets of input data were reserved for testing. The performance metrics of the RF-based models were listed in Table 3. Specifically, F1 scores were 0.857, 0.906, and 0.746, and the AUCs were 0.959, 0.974, and 0.956, for predictions at 36, 37, and 40 weeks PMA on the testing data set, respectively. Overall, the models for 36- and 37-week PMA respiratory outcome predictions had better generalizability compared to the model for predicting the 40-week outcome. A comprehensive list of performance metrics for all three modeling approaches, namely SVM, RF, and ensemble, is available in S1 Table.

**Table 2. Characteristics of infants included in model development and testing.**

| All infants included in the model development and testing (n = 29) | |
|---|---|
| Steroid exposure | |
| Antenatal, ≥ 1 dose | 29 (100%) |
| Postnatal, for BPD prevention | 5 (17.2%) |
| Surfactant | |
| None | 8 (27.6%) |
| ≤ 2 doses | 18 (62.1%) |
| ≥ 3 doses | 3 (10.3%) |
| Comorbidity of prematurity | |
| Intraventricular hemorrhage, any grade | 6 (20.7%) |
| Intraventricular hemorrhage, Grade 3 or 4 | 3 (10.3%) |
| Medical therapy for patent ductus arteriosus | 17 (58.6%) |
| Invasive therapy for patent ductus arteriosus* | 8 (27.6%) |
| Spontaneous intestinal perforation | 1 (3.4%) |
| Necrotizing enterocolitis, Stage 2 or 3 | 2 (6.9%) |
| Respiratory support at postmenstrual age 36 weeks | |
| No support | 8 (27.6%) |
| Low flow nasal cannula | 8 (27.6%) |
| Other ventilatory support | 13 (44.8%) |
| Respiratory support at postmenstrual age 37 weeks | |
| No support | 10 (34.5%) |
| Low flow nasal cannula | 7 (24.1%) |
| Other ventilatory support | 12 (41.4%) |
| Respiratory support at postmenstrual age 40 weeks | |
| No support | 15 (51.7%) |
| Low flow nasal cannula | 9 (31.0%) |
| Other non-invasive ventilatory support | 5 (17.3%) |

*including surgical ligation or coil embolization.

†Inborn if same birth and discharge hospitals; outborn if different birth and discharge hospitals.

BPD: bronchopulmonary dysplasia.

**Table 3. Model performance measures for the indicated outcomes using a random forest algorithm.**

| PMA | 36 weeks | | 37 weeks | | 40 weeks | |
|---|---|---|---|---|---|---|
| Input | 15 probability scores (5 severity categories generated by the BPD Estimator for each of the three surrogate race/ethnicity) | | | | | |
| Positive Outcome | HFNC, CPAP, Invasive respiratory support | | Any respiratory support | | Any respiratory support | |
| Negative Outcome | LFNC, No support | | No support | | No support | |
| Data set | Training | Testing | Training | Testing | Training | Testing |
| Sensitivity | 0.843 | 0.808 | 0.873 | 0.850 | 0.821 | 0.647 |
| Specificity | 0.905 | 0.929 | 0.886 | 0.929 | 0.879 | 0.850 |
| PPV | 0.878 | 0.913 | 0.945 | 0.971 | 0.868 | 0.880 |
| NPV | 0.877 | 0.839 | 0.756 | 0.684 | 0.836 | 0.586 |
| Accuracy (95% CI*) | 0.877 (0.803–0.931) | 0.870 (0.751–0.946) | 0.877 (0.803–0.931) | 0.870 (0.751–0.946) | 0.851 (0.772–0.911) | 0.722 (0.584–0.835) |
| Kappa | 0.751 | 0.739 | 0.725 | 0.698 | 0.701 | 0.455 |
| F1 score | 0.860 | 0.857 | 0.908 | 0.906 | 0.844 | 0.746 |
| ROCAUC (95% CI**) | 0.959 (0.926–0.991) | 0.934 (0.869–1.000) | 0.974 (0.951–0.997) | 0.850 (0.725–0.975) | 0.956 (0.923–0.989) | 0.757 (0.625–0.890) |

PMA: postmenstrual age; HFNC: high-flow nasal cannula; CPAP: continuous positive airway pressure; LFNC: low-flow nasal cannula; PPV: positive predictive value; NPV: negative predictive value; ROCAUC: receiver's operating characteristic area under receiver operating characteristic curve. CI: confidence interval.

*Binomial proportion confidence interval.

**Calculated by the DeLong method.

## Discussion

A recent report using the 2016 national pediatric hospitalization dataset, the Kids' Inpatient Database, showed that Asian, Pacific Islander, and Native Americans as a group had the lowest rate of extreme prematurity when compared to non-Hispanic white, non-Hispanic black, and Hispanic groups. Paradoxically, the report also showed that Asian/Pacific Islander/Native American premature newborns were associated with the highest hospitalization cost, which could be due to a higher rate of morbidities [21]. BPD is one of the major comorbidities of extreme prematurity that may lengthen neonatal hospitalization due to prolonged cardiopulmonary instability. In this report, we developed a machine learning-based solution to address one major limitation of the NICHD BPD Outcome Estimator–the inability to be applied to the Asian population [12]. Respiratory outcome prediction of Asian infants was achieved by using the output from the Estimator with each of the three race/ethnicity surrogates plus the same level of respiratory support as well as the same birth weight and day of life information. Unlike the original Estimator, the models developed in this study were designed to predict two-level categorical respiratory outcomes at various PMA, as we were not able to make predictions for each risk category due to a relatively small sample size. Additionally, the models were not generalizable to Pacific Islanders and Native American because our cohort did not consist of infants from these two racial/ethnic groups.

We chose to use dichotomized outcome measures for simplicity and easy application in the clinical setting. We classified LFNC and no respiratory support as one outcome group at 36 weeks PMA because, from a physiological and developmental standpoint, most of these infants are still developing their oral skills at this age; being able to tolerate LFNC or no support indicates that they will most likely be allowed to receive oral skill training with breast or bottle feeding. The use of LFNC may be due to emerging respiratory reserve and/or residual resolving pulmonary hypertension associated with BPD. At 37 and 40 weeks, we dichotomized the outcome as with or without support needs. Based on the ROCAUC and the F1 scores, the model for predicting respiratory outcomes at 40 weeks PMA showed relatively poor

generalizability, although the AUC for the testing data set was still comparable to the original model used in the BPD Outcome estimator.

In this project, we used a well-established decision tree-based ML technique to extend the usability of the Estimator to the Asian population. While our cohort size was small, the advantage of this study was the ability to set aside a portion of the input data for model testing. Similar to the statistical model behind the Estimator, our model did not take repeated measurements for each infant into consideration. We also chose a dichotomized outcome, as opposed to risk stratification primarily due to small sample size and clinical applicability, as estimation at distinct PMAs longitudinally may be easier to comprehend and communicate in the clinical settings.

We developed a web app for the demonstration of these models. The web app is accessible at https://neostat.shinyapps.io/BPD_Asian/. It requires the probability data for each risk category using White, Black, and Hispanic as race/ethnicity. The prediction output is the probability of having the indicated positive outcome for each PMA. Please note that the prediction models developed in this study used in the web app were based on data and outcomes of our practice and were not thought to be generalizable as they have not gone through rigorous prospective validation, which is a major limitation to this study. This web app was not designed to guide clinical decision-making at the moment. Readers taking care of Asian extremely preterm infants may develop local models based on their own institutional data.

In conclusion, this study suggested an interrelationship between racial/ethnic groups for respiratory outcomes among extremely preterm infants and showed the feasibility of extending the use of the Estimator to the Asian population.

## Supporting information

**S1 Dataset.**
(CSV)

**S1 Table. Model performance comparisons.**
(DOCX)

## Acknowledgments

The authors would like to thank Dr. John B. C. Tan, PhD of Huckleberry Labs for critically reviewing the manuscript.

## Author Contributions

**Conceptualization:** Fu-Sheng Chou.

**Data curation:** Japmeet Sandhu, Fu-Sheng Chou.

**Formal analysis:** Monalisa Patel, Fu-Sheng Chou.

**Writing – original draft:** Monalisa Patel, Japmeet Sandhu, Fu-Sheng Chou.

**Writing – review & editing:** Monalisa Patel, Fu-Sheng Chou.

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
