## [Decision Letter · Decision Letter 0]

9 Dec 2021

PONE-D-21-16900Developing a machine learning-based tool to extend the usability of the NICHD BPD Outcome Estimator to the Asian populationPLOS ONE

Dear Dr. Chou,

Thank you for submitting your manuscript to PLOS ONE. After careful consideration, we feel that it has merit but does not fully meet PLOS ONE’s publication criteria as it currently stands. Therefore, we invite you to submit a revised version of the manuscript that addresses the points raised during the review process.

We look forward to receiving your revised manuscript.

Kind regards,

Jianhong Zhou

Associate Editor

PLOS ONE

Journal Requirements:

2. Please include your full ethics statement in the manuscript Methods.

Reviewers' comments:

Reviewer's Responses to Questions

**Comments to the Author**

1. Is the manuscript technically sound, and do the data support the conclusions?

Reviewer #1: Partly

Reviewer #2: Partly

2. Has the statistical analysis been performed appropriately and rigorously? 

Reviewer #1: I Don't Know

Reviewer #2: Yes

3. Have the authors made all data underlying the findings in their manuscript fully available?

Reviewer #1: Yes

Reviewer #2: Yes

4. Is the manuscript presented in an intelligible fashion and written in standard English?

Reviewer #1: Yes

Reviewer #2: Yes

5. Review Comments to the Author

Reviewer #1: The article "Developing a machine learning-based tool to extend the usability of the NICHD Outcome Calculator to the Asian population," by Sandhu et al is an interesting application of the BPD Estimator to a local population of infants of Asian descent. Given disparities in outcomes based on racial demographics in infants with BPD, this idea is novel and may be useful in counseling families of Asian descent who have premature infants at risk for BPD. The developed model seems to have good predictive capability at 36 and 37 weeks', but there are some general concerns the authors should address.

1) Abstract: The objective of the study seems to be to apply the Estimator to an Asian demographic and evaluate its performance. However, the hypothesis and goal is unclear and should be revised.

2) Introduction: In general, when making statements about the incidence of BPD or inflammation as an etiology of BPD etc, references should be stated. Many BPD experts would argue about inflammation as being the most common cause of BPD, as it is largely considered to be of multi-factorial origin. Additionally, the authors claim the most significant intervention in preventing mortality and the incidence of BPD are systemic steroids. Although there is some evidence to support this claim, there was also a large NICHD study (Watterburg et al) that does not support this claim. On the whole, the authors should focus the introduction more on what is known about predictors of BPD in an Asian population, and the overall incidence in this group, to support their hypothesis more clearly.

Methods: There should be a statement regarding approval by an Institutional Review Board.

The methods are a little unclear. It seems that the Asian patient population was entered into the BPD Estimator using surrogate races, and that the prediction of the calculator was then compared to the observed outcomes at 36, 37 and 40 weeks'? The authors should clarify the approach some in the Methods section. How Asian descent was determined is an important factor in this study.

Results: The validity of the model is enhanced by having a training and testing set. It could be further strengthened by testing it on an independent data set outside the institution if that is available. It may be interesting to evaluate the performance of the model based on the surrogate race/ethnicity used as input. For example, was there one race/ethnicity that is already in the calculator that performed well in the Asian descent population?

Discussion: The tool included in the Discussion is useful. The authors should address how to apply this in a clinical setting. They should also include confidence limits around the point estimate.

Reviewer #2: This interesting study assessed whether outcome prediction for Asian preterm born infants is possible using information hidden in prediction results for the other major racial groups in het NICHD BPD Outcome Estimator. Prediction of BPD development is an important topic, and investigating whether the risk can be predicted in infants with other descent then is used now in the Estimator is interesting.

After reading the manuscript, I have some addition questions and comments.

1. In the methods section the authors described a retrospective study with prospective data collection, although at the end they describe that retrospective data collection was approved by the institutional review board. So it is unclear whether the data was collected prospectively or retrospectively? And where any exclusion criteria used for this study?

2. In the methods section the authors describe the part of machine learning-bases model development. First of all, good that the authors have performed internal validation (train and test set) as part of model development, however no information is presented whether the authors performed shrinkage of the predictor weights or regression coefficients. In addition, it is not clear which specific variables were used for model development. Are only the five severity categories are used? Or also the demographic data on GA, BW, respiratory support etc. And in case these demographic data is used, from which timepoint are these?

3. In the methods section no information is presented about the usage of any guidelines for development of prediction models? And no information is presented about how the authors dealed with missing data?

4. In the results section the authors described the inclusion of 842 premature newborns with a gestational age equal or less to 30 weeks and 6 days, although the inclusion criteria for this study described in the methods section, stated GA 30 weeks or less.

5. There is no information about how many infants developed BPD in the study cohort.

6. Table 3 missed information about which variables are included in the model.

7. In the results section information about the comparison of the outcomes from the machine-based learning models and the outcomes from the NICHD BPD Outcome Estimator is missing. In my opinion this is necessary to underlie the conclusion that extending the use of the Estimator to Asian population is feasible. In addition, no information is presented regarding the final predictor weights?

8. It can be suggested to add a calibration plot of model performances from both the train and test model in the results section.

9. From the discussion, no clear conclusion is presented. Only in the Abstract, a clear conclusion is presented.

6. PLOS authors have the option to publish the peer review history of their article (what does this mean?). If published, this will include your full peer review and any attached files.

Reviewer #1: No

Reviewer #2: **Yes: **Michelle Romijn

---

## [Author Response · Author response to Decision Letter 0]

20 Jan 2022

Reviewer #1: The article "Developing a machine learning-based tool to extend the usability of the NICHD Outcome Calculator to the Asian population," by Sandhu et al is an interesting application of the BPD Estimator to a local population of infants of Asian descent. Given disparities in outcomes based on racial demographics in infants with BPD, this idea is novel and may be useful in counseling families of Asian descent who have premature infants at risk for BPD. The developed model seems to have good predictive capability at 36 and 37 weeks', but there are some general concerns the authors should address.

1) Abstract: The objective of the study seems to be to apply the Estimator to an Asian demographic and evaluate its performance. However, the hypothesis and goal is unclear and should be revised.

Response: We thank the reviewer for pointing out the deficiency in the abstract. We have revised it to bring more clearness to the hypothesis and goal of the study. Please refer to the revised manuscript.

2) Introduction: In general, when making statements about the incidence of BPD or inflammation as an etiology of BPD etc, references should be stated. Many BPD experts would argue about inflammation as being the most common cause of BPD, as it is largely considered to be of multi-factorial origin. 

Response: We thank the reviewer for pointing out that BPD is a multifactorial disease, which we agree. We have updated this part of the introduction (Paragraphs 1 & 2) and have also included supporting references.

Additionally, the authors claim the most significant intervention in preventing mortality and the incidence of BPD are systemic steroids. Although there is some evidence to support this claim, there was also a large NICHD study (Watterburg et al) that does not support this claim. On the whole, the authors should focus the introduction more on what is known about predictors of BPD in an Asian population, and the overall incidence in this group, to support their hypothesis more clearly.

Response: We have revised the Introduction section to put more emphasis on the lack of a prediction tool for Asian population. Please refer to Page 5 of the revised manuscript.

3) Methods: There should be a statement regarding approval by an Institutional Review Board.

Response: We thank the reviewer for pointing out the deficiency. We have added the statement to the Methods section. Please refer to the revised manuscript. Please refer to Page 6 of the revised manuscript.

The methods are a little unclear. It seems that the Asian patient population was entered into the BPD Estimator using surrogate races, and that the prediction of the calculator was then compared to the observed outcomes at 36, 37 and 40 weeks'? The authors should clarify the approach some in the Methods section. How Asian descent was determined is an important factor in this study.

Response: The reviewer’s understanding of the methodology was correct. We agree that we could have been clearer in describing our methods. The identification of the Asian descent is based on the facesheet available in the electronic medical records. We have edited the methods section to provide more clarity. Please refer to Pages 6 & 7 of the revised manuscript.

4) Results: The validity of the model is enhanced by having a training and testing set. It could be further strengthened by testing it on an independent data set outside the institution if that is available. It may be interesting to evaluate the performance of the model based on the surrogate race/ethnicity used as input. For example, was there one race/ethnicity that is already in the calculator that performed well in the Asian descent population?

Response: Publication of this project will strength our case of obtaining funding and access to a multi-center de-identified research data warehouse to upscale the study. We did not specifically examine the data to answer the question posted by the reviewer. We felt that, since race/ethnicity is social determinant, it would be very difficult to interpret the findings and to provide a sound biological explanation if one specific race/ethnicity performs well in predicting BPD outcome for Asian infants. Please forgive us for not being able to provide a more satisfying answer given the sensitive nature of the race/ethnicity topic.

Discussion: The tool included in the Discussion is useful. The authors should address how to apply this in a clinical setting. They should also include confidence limits around the point estimate.

Response: We added a disclaimer to the web app and also revised the Discuss paragraph to emphasize that the web app developed for this project is meant for demonstration purpose and should not be used to guide clinical decision-making due to the lack of external validation. We added the 95% confidence intervals for accuracy. To calculate receiver’s operating characteristic area under the curve (ROCAUC), we went back to use probability to recalculate ROCAUC (instead of the AUC based on one single cutoff value as shown previously). The generalizability for CGA 40 week respiratory outcome is still much poorer compared to that for CGA 36 and 37 weeks. Please refer to the revised Table 3.

Reviewer #2: This interesting study assessed whether outcome prediction for Asian preterm born infants is possible using information hidden in prediction results for the other major racial groups in the NICHD BPD Outcome Estimator. Prediction of BPD development is an important topic, and investigating whether the risk can be predicted in infants with other descent then is used now in the Estimator is interesting.

After reading the manuscript, I have some additional questions and comments.

1. In the methods section the authors described a retrospective study with prospective data collection, although at the end they describe that retrospective data collection was approved by the institutional review board. So it is unclear whether the data was collected prospectively or retrospectively?

Response: Thank you for pointing this out. We conducted a retrospective study. Data were collected in retrospect. We have updated the manuscript. 

Were there any exclusion criteria used for this study?

Response: The exclusion criteria were those that did not meet GA and birth weight criteria, and those that were not designated as Asian descent on the facesheet. This was a single-center pilot study so we wanted to be as inclusive as possible. We have updated and clarified our inclusion and exclusion criteria in our methods. 

2. In the methods section the authors describe the part of machine learning-bases model development. First of all, good that the authors have performed internal validation (train and test set) as part of model development, however no information is presented whether the authors performed shrinkage of the predictor weights or regression coefficients.

Response: We thank the reviewer for the comment. We believe the reviewer is referring to the elastic net or other related linear algorithms, where coefficient weights were assessed by the machine. The random forest is a decision tree-based algorithm, and the support vector machine is a distance-based approach in a hyperspace. Assessment of shrinkage of the predictor weights, in our understanding, is not applicable to these algorithms.

In addition, it is not clear which specific variables were used for model development. Are only the five severity categories are used? Or also the demographic data on GA, BW, respiratory support etc. And in case these demographic data is used, from which timepoint are these?

Response: The assumption for the model development is that the five probability scores for all five severity categories for each surrogate race/ethnicity assignment already incorporates the demographic and respiratory support information in them. Therefore, only probability scores (a total of 15 as a result of 5 severity categories with 3 surrogate race/ethnicity) were incorporated as features for model development. We have revised the manuscript for better clarity. Please refer to Page 7 of the revised manuscript.

3. In the methods section no information is presented about the usage of any guidelines for development of prediction models? 

Response: Thank you for pointing this out. We followed the TRIPOD guidelines checklist for prediction model development and validation. We have updated Methods to include this detail. 

And no information is presented about how the authors dealt with missing data?

Response: The probability scores generated from the BPD estimator were used as predictors for model development, therefore there was no missing data. If the birth weight fell out of the range allowed in the Estimator based on the other demographic data (but still within 500-1,250g), we used the closest weight. Babies that did not meet our inclusion criteria for gestational age and babies that did not have documented Asian race, were excluded. 

4. In the results section the authors described the inclusion of 842 premature newborns with a gestational age equal or less to 30 weeks and 6 days, although the inclusion criteria for this study described in the methods section, stated GA 30 weeks or less.

Response: Thank you for pointing this out. We have corrected our description in Methods and Results to consistently state GA 30 weeks or less.

5. There is no information about how many infants developed BPD in the study cohort.

Response: We have updated our results section to further clarify these details. BPD is generally diagnosed when an infant requires continued respiratory support at 36 weeks CGA. We provided breakdown of types of respiratory support at 36, 37, and 40 weeks CGA by partially following the BPD grading system per Jensen et al (2019). Please refer to Table 2. 

6. Table 3 missed information about which variables are included in the model.

Response: We only used the probability scores as input. We have updated the table. Please refer to Page 14 of the revised manuscript.

7. In the results section information about the comparison of the outcomes from the machine-based learning models and the outcomes from the NICHD BPD Outcome Estimator is missing. In my opinion this is necessary to underlie the conclusion that extending the use of the Estimator to Asian population is feasible. 

Response: It is not possible to compare outcomes from the Estimator to the machine-based learning models as the NICHD BPD Outcome Estimator does not apply to infants of Asian descent, hence we developed this pilot project to address this deficiency of the Estimator.

In addition, no information is presented regarding the final predictor weights?

Response: In this single-center project with relatively small sample size, we felt that delivery of variance importance scores to the readers will create confusion and the relative importance of the probability scores cannot be explained in biological terms. We would like to request permission from the reviewer to not present predictor weights unless a reader requests the information in written correspondence after publication.

8. It can be suggested to add a calibration plot of model performances from both the train and test model in the results section.

Response: It is our understanding that calibration curves require sufficiently large samples, hence we didn't find it particularly useful for this single-center pilot study.

9. From the discussion, no clear conclusion is presented. Only in the Abstract, a clear conclusion is presented.

Response: We added a conclusion statement to the end of Discussion. Please refer to the revised manuscript.

---

## [Decision Letter · Decision Letter 1]

2 May 2022

PONE-D-21-16900R1Developing a machine learning-based tool to extend the usability of the NICHD BPD Outcome Estimator to the Asian populationPLOS ONE

Dear Dr. Chou,

Thank you for submitting your manuscript to PLOS ONE. After careful consideration, we feel that it has merit but does not fully meet PLOS ONE’s publication criteria as it currently stands. Therefore, we invite you to submit a revised version of the manuscript that addresses the points raised during the review process.

We look forward to receiving your revised manuscript.

Kind regards,

Jose Palma, Ph.D.

Academic Editor

PLOS ONE

Journal Requirements:

Reviewers' comments:

Reviewer's Responses to Questions

**Comments to the Author**

1. If the authors have adequately addressed your comments raised in a previous round of review and you feel that this manuscript is now acceptable for publication, you may indicate that here to bypass the “Comments to the Author” section, enter your conflict of interest statement in the “Confidential to Editor” section, and submit your "Accept" recommendation.

Reviewer #1: (No Response)

2. Is the manuscript technically sound, and do the data support the conclusions?

Reviewer #1: Yes

3. Has the statistical analysis been performed appropriately and rigorously? 

Reviewer #1: Yes

4. Have the authors made all data underlying the findings in their manuscript fully available?

Reviewer #1: Yes

5. Is the manuscript presented in an intelligible fashion and written in standard English?

Reviewer #1: Yes

6. Review Comments to the Author

Reviewer #1: The authors have done a good job at addressing most Reviewer comments. However, there are still a few minor points that need to be addressed.

Abstract:

1. The sentence that the Estimator does not apply to infants of Asian descent should be modified to say that is not known if it can be applied (I think this is the hypothesis)?

2. Is the hypothesis that the respiratory outcomes are interconnected between respiratory groups, or that the Estimator can be safely used to predict outcomes in infants of Asian descent? I think it is the latter? There is a grammatical error in this sentence.

Introduction

1. Neonatology on Page 4 does not need to be capitalized.

2. There are grammatical errors in the Introduction.

3. The hypothesis needs to be clarified (as stated for the Abstract).

Results

1. It may be important to know how many infants had missing race information on the facesheet as that may introduce some sampling bias.

2. On page 10, the word “had” should be removed from the sentence “At 37 and 40 weeks, CGA …”

Discussion

1. The national pediatric hospitalization dataset should be capitalized if it is a specific dataset that is being named.

2. The statement “The use of LFNC indicates…residual resolving pulmonary hypertension associated with BPD” is misleading. Level of respiratory support alone is not used to determine PH risk.

3. The authors should carefully proofread the manuscript and correct grammatical errors prior to acceptance.

7. PLOS authors have the option to publish the peer review history of their article (what does this mean?). If published, this will include your full peer review and any attached files.

Reviewer #1: No

---

## [Author Response · Author response to Decision Letter 1]

5 May 2022

Abstract:

1. The sentence that the Estimator does not apply to infants of Asian descent should be modified to say that is not known if it can be applied (I think this is the hypothesis)?

Sorry for the confusion. The Estimator really cannot be applied to infants of Asian descent because of the limitation in the choices in the Race/Ethnicity pull-down menu. We changed the sentence to “However, the estimator does not have an option in its pull-down menu for infants of Asian descent.”. Please refer to the revised manuscript.

2. Is the hypothesis that the respiratory outcomes are interconnected between respiratory groups, or that the Estimator can be safely used to predict outcomes in infants of Asian descent? I think it is the latter? There is a grammatical error in this sentence.

The grammatical error has been corrected. Thank you for pointing this out. 

Acknowledging that prediction model development projects are not traditional hypothesis-driven research studies, we tried to rephrase the presumption in model design into a hypothesis but could not do a good job. Thank you for wording it so elegantly for us. We hope you don’t mind us using both of your statements in our revised manuscript.

Introduction

1. Neonatology on Page 4 does not need to be capitalized.

Corrected.

2. There are grammatical errors in the Introduction.

We have revised parts of the Introduction section and have grammar checked by a native English speaker who is also a biomedical researcher. Please kindly review the revised manuscript and let us know if more work is still needed.

3. The hypothesis needs to be clarified (as stated for the Abstract).

Updated in the revised manuscript.

Results

1. It may be important to know how many infants had missing race information on the facesheet as that may introduce some sampling bias.

This has been updated in the revised manuscript.

2. On page 10, the word “had” should be removed from the sentence “At 37 and 40 weeks, CGA …”

Corrected. Thank you for pointing this out.

Discussion

1. The national pediatric hospitalization dataset should be capitalized if it is a specific dataset that is being named.

The name of the database is Kids’ Inpatient Database. It has been added to the revised manuscript and capitalized.

2. The statement “The use of LFNC indicates…residual resolving pulmonary hypertension associated with BPD” is misleading. Level of respiratory support alone is not used to determine PH risk.

We changed “indicates” to “may be associated with” in the revised manuscript. Please assist in assessing the adequacy of such wording.

3. The authors should carefully proofread the manuscript and correct grammatical errors prior to acceptance.

We have sought medical writing expert to assist in proofreading. Please kindly review the revised manuscript and advise whether more work is still needed.

---

## [Decision Letter · Decision Letter 2]

26 Jul 2022

Developing a machine learning-based tool to extend the usability of the NICHD BPD Outcome Estimator to the Asian population

PONE-D-21-16900R2

Dear Dr. Chou,

We’re pleased to inform you that your manuscript has been judged scientifically suitable for publication and will be formally accepted for publication once it meets all outstanding technical requirements.

Kind regards,

Jose Palma, Ph.D.

Academic Editor

PLOS ONE

Additional Editor Comments (optional):

Reviewers' comments:

Reviewer's Responses to Questions

**Comments to the Author**

1. If the authors have adequately addressed your comments raised in a previous round of review and you feel that this manuscript is now acceptable for publication, you may indicate that here to bypass the “Comments to the Author” section, enter your conflict of interest statement in the “Confidential to Editor” section, and submit your "Accept" recommendation.

Reviewer #3: All comments have been addressed

2. Is the manuscript technically sound, and do the data support the conclusions?

Reviewer #3: Yes

3. Has the statistical analysis been performed appropriately and rigorously? 

Reviewer #3: Yes

4. Have the authors made all data underlying the findings in their manuscript fully available?

Reviewer #3: Yes

5. Is the manuscript presented in an intelligible fashion and written in standard English?

Reviewer #3: Yes

6. Review Comments to the Author

Reviewer #3: The issues raised by the reviewer have been adequately addressed and the article is now ready for publication.

7. PLOS authors have the option to publish the peer review history of their article (what does this mean?). If published, this will include your full peer review and any attached files.

Reviewer #3: No

---

## [Editor Report · Acceptance letter]

8 Sep 2022

PONE-D-21-16900R2 

Developing a machine learning-based tool to extend the usability of the NICHD BPD Outcome Estimator to the Asian population 

Dear Dr. Chou:

I'm pleased to inform you that your manuscript has been deemed suitable for publication in PLOS ONE. Congratulations! Your manuscript is now with our production department. 

Kind regards, 

on behalf of

Mr. Jose Palma 

Academic Editor

PLOS ONE